# Organizational debt—Roadblock to agility in software engineering: Exploring an emerging concept and future research for software excellence

Osama Al-Baik[1]*, Mwaffaq Abu Alhija[2], Hikmat Abdeljaber[3], Muhammad Ovais Ahmad[4]

1 Department of Software Engineering, Princess Sumaya University for Technology, Amman, Jordan, 2 Department of Computer Science, Al-Ahliyya Amman University, Amman, Jordan, 3 Department of Computer Science, Applied Science Private University, Amman, Jordan, 4 Department of Computer Science, Karlstad University, Karlstad, Sweden

* o.albaik@psut.edu.jo

## Abstract

In software engineering, organizational debt (OD) is a crucial but little-researched phenomena. OD refers to the accumulation of outdated structures, policies, and processes that hinder an organization's advancement and adaptability. This multivocal literature review (MLR) synthesizes insights from software practitioners to elucidate OD causes, consequences, identification, and mitigation approaches that is considered a first step in illuminating the OD for software practitioners. After a thorough search, nine peer-reviewed articles and twenty-two recent blog posts on OD were included, indicating an emerging topic. Through inductive thematic analysis, four key topics emerged: definitions, causes like poorly managed change and siloed efforts, effects such as reduced innovation and agility, and mitigation strategies including agile principles, decentralized decision-making, and leveraging staff insights. While relying partly on non-peer-reviewed sources raises validity concerns, the review still provides a holistic and practical understanding of OD dynamics and complexities grounded in diverse perspectives. Further empirical research across diverse organizations would strengthen these preliminary findings. Effective OD management necessitates collaboration between academia and industry, considering technical debt (TD) best practices while tailoring interventions to OD's distinct socio-technical characteristics.

## 1 Introduction

Software development is a dynamic field with many moving parts that result from the complicated interactions of many stakeholders, each of whom brings unique priorities and points of view to the table [1]. The complex character of software creation processes is further compounded by the unpredictable nature of human behavior in the development setting, as well as the particular problems posed by the different requirements of each project and the diverse backgrounds of all parties involved [2, 3]. Moreover, the relentless evolution of technologies and methodologies, interacting in unpredictable ways with organizational cultures and structures, introduces an additional layer of intricacy [4, 5].

**Data Availability Statement:** All relevant data are within the manuscript and its Supporting information files.

**Funding:** This work was supported by the KK-stiftelsen, Sweden, through the NODLA project (20200253 to M.O.A) and Helge Ax: son Johnsons Stiftelse.

**Competing interests:** The authors have declared that no competing interests exist.

The adoption of agile methodologies, emphasizing iterative development, constant collaboration with customers, and continual process improvements, represents an acknowledgement of the complexities inherent in contemporary software engineering [5, 6]. However, a critical concern emerges when development teams are treated as interchangeable resources subject to frequent reassignment between projects, disrupting team cohesion and knowledge transfer, potentially compromising productivity and quality [2]. Proponents of agile highlight the need for stable teams to realize the full benefits of iterative approaches and cross-functional collaboration [4].

While TD has garnered significant research attention, encompassing efforts, tools, types, management strategies, architectural aspects, agile development, and prioritization [7–9], there exists a noticeable paucity of exploration into the realm of OD. This paper aims to illuminate the phenomenon of OD through a MLR, synthesizing insights from software practitioners to uncover causes, consequences, identification, and mitigation approaches related to OD.

We conducted a search of Google Scholar in July 2023 with the keywords "organizational debt" AND "software". The search returned 120 results that were screened based on inclusion criteria of relevance to software engineering and organizational debt. A total of 22 sources met the criteria and were analyzed thematically to identify key topics related to causes, consequences, and mitigation strategies. As there were no academic articles on the OD and software, and in attempt to strengthen the research findings, another search was conducted in June 2024 with the keywords "organizational debt" AND (management OR "organizational behavior" OR "information systems"). This search resulted in including nine articles.

The subsequent sections of this pare are organized as follows, section 2 presents the background, section 3 explains the research methodology, section 4 reports the results, while section 5 discusses the validity assessment, section 6 dives into an in-depth discussion, and finally section 7 provides conclusions gleaned from the rigorous literature analysis.

## 2 Background

The software development process is a multifaceted and intricate undertaking characterized by the complex interplay of numerous stakeholders, each contributing to the dynamism of the environment. The unpredictability of human behavior within this context, coupled with the unique challenges presented by diverse project requirements and the distinct backgrounds of the individuals involved, further amplifies the complexity of the software development landscape [2]. The continuous advancement of technology and organizational procedures adds complexity, as these factors interact in unforeseen and frequently mutually beneficial ways.

Implementation of dynamic Agile and lean methodologies, characterized by iterative value delivery to clients and the elimination of wasteful practices that impact productivity and quality, acknowledge this complex nature [4, 6, 10, 11]. However, a significant issue occurs when individuals are perceived as replaceable "resources" who can be often reassigned to different positions. The empirical research clearly demonstrates that this technique is completely unproductive and has a detrimental impact on team relations for both the sender and the recipient. This poses a challenge to the transmission of information and, at times, even puts overall productivity and quality at risk.

Another challenge is enforcing a rigidly inflexible process, which sometimes gives managers certain bureaucratic complacencies between them, erroneously clinging to the belief that this very same process by itself could be some cure for all possible worse-case scenarios ever. Inevitably, such rigidity leads to blaming the team for deviations from a rigidly prescribed process when unexpected challenges ultimately arise a cultural blockade that frustrates continuous improvement [2].

The prevalent "Just get it done!" attitude though arising out of the desire for rapid development, works as a main source in software heterodony. As Cunningham pointed out in 1992 [12], there is a degree of borrowing enclosing code for the first time that resembles taking out a loan, and while little debt may fasten development, this must be promptly paid back with refactoring. There is no equivalent nefariousness of the TD, as each and every minute spent with a code that otherwise contains might be aggregated into interest on this type of financial loan. This definition was furthered by Avgeriou et al. [7] to include a range of TD artifacts, consisting of architectural debt and code as well as test-type debt.

TD has been studied in extreme depth from various perspectives, including efforts, tools, types, management strategies, architectural aspects, and agile development or prioritization [8, 9, 13–18]. For example, 78 causes and 66 effects of TD have been identified by Rios et al. [15], considering a series of reasons, from tight timeframes to poor documentation or emphasis on high volumes over quality. Avgeriou et al. [16] made comparisons of tools for quantifying TD, noting the influence on maintainability, quality, project expenses, financial losses, code refinement, team overload, and dissatisfaction of the stakeholders. In particular, the study highlights the importance of management principles that can help manage architectural debt effectively, especially in a scenario where agile software development is involved.

In addition to TD, the notion of Non-Technical Debt (NTD) has been introduced to encompass various organizational debts that are not technical in nature [14, 19]. These debts include process, social, human, and cultural in nature. NTD arises from imprudent decision-making that prioritizes immediate benefits over long-term sustainability, thereby impeding numerous software development endeavors and influencing the outcome of projects. In consideration of the literature, the principal NTD types are examined and outlined below.

*Process debt*, a type of suboptimal conduct that provides immediate advantages but has enduring repercussions, may result from conducting these meetings solely for the purpose of updating statuses, thereby restricting their complete potential. This is largely attributable to a lack of process competencies, a gap resulting from a subtle deviation from the norm, and external influences such as technological tools and trends [19]. *People debt* refers to development matters such as owed specialization on a limited number of staff accruing from late training and recruitment. Those factors leading to people debt are suggested a lack of knowledge, experience, devotion, and psychological safety; inappropriate management decisions acceptance issues among personnel or diminutive developer reasonable confidence [19]. *Cultural debt.* Technical decisions that lead to delinquency to culture may consume a significant portion or the entire culture of an organization. This can potentially result in the division of teams, deterioration of communication practices, and reduced effectiveness of leadership [14]. Social debt may be detrimental to the suboptimal development environment of numerous software development communities, as evidenced by gender bias resulting from communication and collaboration constraints [19].

Notably, with the advent of novel hybrid work policies in contemporary organizational environments, managers may encounter the peril of OD. Policies of this nature, which are instrumental in shaping the development of emerging norms among employees, necessitate meticulous scrutiny to avert unfavorable consequences. Liu et al. [20] identified and analyzed the impact of twenty-three mechanisms, which they categorized into eight coordination domains and implemented in shared mental models, team coordination, cohesion, and learning. The software development process is inherently complicated and requires a multidimensional approach that considers the specific characteristics of each project's participants and their changing technological positions during organizational process transitions. By highlighting the importance of delivering value, waste reduction, and adaptability, the combination of Agile and Lean approaches can provide a solution to these difficulties [4, 6, 10, 11].

## 3 Research method

One of the tools used to handle various issues that arise in the field of software engineering is the MLR, This study employed MLR approach, which combines the analysis of practitioner literature (grey literature) with academic sources to provide a comprehensive understanding of the investigated phenomenon [21]. This method includes a broad spectrum of written sources that are easily accessible, considers several authors' perspectives, and even touches upon non-scientific subjects. For MLR to be applied effectively, it is important for a specific research question to be formulated since MLR involves manifold points of view. Our study aimed to achieve a deep understanding of how software practitioners articulate the concept of 'organizational debt'. In an attempt to ensure rigorous research, the authors used the Preferred Reporting Items for Systematic reviews and Meta-Analyses (PRISMA), to report the research items (see [S1 Checklist]).

### 3.1 Initial grey literature search

The deliberate source identification approach was used in this MLR as Garousi and Mantyla suggested [21]. The Google search engine was used as a tool of initial search instead of traditional scholarly databases like SpringerLink, ACM, and IEEE eXplore. This conviction was based on the fact that the selected topic for investigation, OD, is quite new and there are few sources of the relevant literature about this issue. The search capabilities of Google made it possible to efficiently search for the grey literatures. In July 2023, we did a literature search using a meticulously designed query ("organisational debt" OR "organizational debt" OR "OD" AND "software") to guarantee a thorough scope.

The inclusion of the term "software" in the search string was deliberate, encompassing studies related to software, software development, software engineering, or software-intensive products, services, and systems. The dual inclusion of ("OD" OR "organizational debt") aimed to encompass all pertinent literature on OD.

A total of 120 sources from the initial search results were scrutinized, aligning with common practices in grey literature research, where researchers often consider the first 50 search results [22]. Each link was meticulously examined, and relevant information was documented. Exclusions were made for sources not in English, videos, advertisements, catalogues, duplicates, research profiles, or those outside the realm of software engineering. Ultimately, twenty-two blog posts from the selected 120 sources were deemed pertinent and are duly cited in the reference list from [23–44], inclusive. Notably, the search strategy did not yield any academic relevant articles. Out of curiosity, we conducted a manual search of Scopus, Web of Science, PubMed, SpringerLink, ACM, and IEEE eXplore. Only one result was obtained, and it was co-authored by one of the co-authors of this study.

### 3.2 Academic literature search

To address the limitation of relying solely on practitioner perspectives and enhance the theoretical grounding of the findings, a comprehensive search for academic literature was conducted. The following databases were searched: Scopus, Web of Science, ABI/INFORM, JSTOR, and Google Scholar.

We conduct a thorough search for relevant academic literature on OD from related disciplines, such as management, organizational behavior, and information systems. This will broaden our data corpus and provide a more comprehensive view of the phenomenon. The search strings used were:

("organizational debt" OR "organisational debt") AND (management OR "organizational behavior" OR "information systems")

The searches were limited to peer-reviewed journal articles and conference proceedings published in English between 2000 and 2023. The initial search yielded 387 results. The following inclusion criteria were applied to the academic literature search results:

1. Empirical studies or conceptual/theoretical papers explicitly addressing organizational debt or closely related concepts (e.g., management debt, organizational inertia, organizational agility).

2. Studies focusing on the causes, consequences, or mitigation strategies related to organizational debt or similar phenomena.

3. Studies from the domains of management, organizational behavior, information systems, or related disciplines.

The exclusion criteria were:

1. Papers not published in English.

2. Papers not available in full text.

3. Papers focusing solely on technical debt without addressing organizational or non-technical debt aspects.

After applying the selection criteria, a total of 9 academic papers were included in the final analysis and are duly cited in the reference list from [45–53], in addition to the 22 blog posts [23–44] from the initial grey literature search [S1 Checklist].

## 3.3 Data analysis

The subsequent data analysis phase employed thematic analysis techniques, focusing on four emergent themes: OD definitions, OD causes, consequences, and mitigation strategies. Unlike a deductive approach with pre-defined themes or codes, our analysis embraced an inductive stance, allowing themes related to causes, effects, and mitigation strategies to surface organically from the data. The utilization of this open analysis methodology amplifies the number and profundity of ideas derived from the literature.

Findings from both the practitioner literature and academic sources were synthesized and integrated within each theme, facilitating a comprehensive understanding of the phenomenon from diverse perspectives. By combining the analysis of grey literature with academic sources, this MLR provides a holistic and multi-faceted exploration of organizational debt, grounded in both practical insights and theoretical foundations. The findings of our MLR analysis are presented in Section 4.

## 4 Results

In this section, we conduct a comprehensive examination of the results obtained from the MLR. The primary objective is to examine the academic literature and progression of grey literature publications and gather diverse explanations of OD. Furthermore, we will investigate the fundamental causes and outcomes of OD, and subsequently evaluate strategies for detecting and reducing it.

### 4.1 Organizational debt concept in related fields

Based on the analysis of the identified academic literature, we have synthesized the key findings related to organizational debt (OD) causes, consequences, and mitigation strategies.

### 4.1.1 Causes of organizational debt.

1. Outdated organizational structures and policies: Failure to adapt structures, processes, and policies to changing business environments and market conditions can lead to the accumulation of OD [45, 48, 49].

2. Short-term decision-making: Prioritizing short-term gains over long-term sustainability and strategic objectives can result in compromises that contribute to OD [47, 48].

3. Lack of organizational agility: Rigid hierarchies, siloed operations, and resistance to change hinder an organization's ability to respond effectively to emerging challenges and opportunities, leading to OD accumulation [46, 51].

4. Inefficient coordination and collaboration: Inadequate mechanisms for cross-functional coordination and knowledge sharing can impede productivity and innovation, contributing to OD [46, 52].

5. Technology-related stressors: Overreliance on technology, inadequate technology governance, and technostress can negatively impact employee productivity and well-being, leading to OD [50, 53].

### 4.1.2 Consequences of organizational debt.

1. Reduced productivity and efficiency: OD can result in suboptimal processes, redundancies, and inefficiencies, negatively impacting overall organizational productivity [45, 48, 52].

2. Diminished innovation and competitiveness: Outdated structures, policies, and processes can stifle creativity, hinder adaptability, and undermine an organization's ability to innovate and remain competitive [47, 49].

3. Employee disengagement and turnover: A dysfunctional work environment, lack of empowerment, and technostress can lead to decreased employee engagement, job satisfaction, and increased turnover [54, 55].

4. Financial implications: OD can result in increased operating costs, missed opportunities, and potential financial losses due to inefficiencies and lack of competitiveness [45, 51]

5. Reputational damage: Failure to address OD and its consequences can harm an organization's reputation, customer satisfaction, and stakeholder trust [49].

### 4.1.3 Mitigation strategies for organizational debt.

1. Continuous organizational assessment and adaptation: Regular audits, employee feedback, and performance monitoring can help identify sources of OD, enabling proactive adjustments to structures, processes, and policies [48, 49].

2. Fostering organizational agility and flexibility: Implementing agile methodologies, decentralized decision-making, and cross-functional collaboration can enhance an organization's ability to respond to change and mitigate OD accumulation [46, 51].

3. Strategic alignment and long-term planning: Ensuring alignment between organizational structures, processes, and strategic objectives, while considering long-term implications of decisions, can prevent OD accumulation [47, 48]

4. Technology governance and digital transformation: Implementing effective technology governance, digital upskilling initiatives, and addressing technostress can mitigate technology-related OD drivers [50, 53]

5. Empowering employees and promoting a learning culture: Encouraging employee autonomy, continuous learning, and innovation can foster an engaged workforce and support OD mitigation efforts [49, 50].

This summary highlights the multifaceted nature of OD, encompassing structural, cultural, technological, and human factors. Effective OD management requires a holistic approach that addresses the underlying causes, mitigates the negative consequences, and promotes organizational adaptability, innovation, and employee well-being.

## 4.2 Trend and definition of organizational debt concept

The MLR systematically assembled twenty-two blog postings centered on OD, authored by seasoned professionals in the software engineering field. A clear pattern emerges as six out of 13 articles were written in the past couple of years 2020–2023, highlighting the increasing discussion surrounding the OD phenomenon. The conceptual roots of OD trace back to 2015 when Steve Blank extended the metaphor of TD and characterized OD as "worse" [35].

Notably, this idea finds antecedents in Ben Horowitz's conceptualization of "management debt" dating back to 2012 [34]. Subsequent contributors like Dignan broadened the scope, asserting that OD is not confined to start-ups but holds significance on a broader organizational scale [32]. The MLR results offer a plethora of OD definitions, reflecting the nuanced perspectives of software industry professionals, Table 1 below shows a summary of these definitions:

The literature review revealed 13 distinct definitions of organisational debt proposed by various practitioners and experts. While differing in precise wording and focus, several core concepts recur across these definitions. Organisational debt refers to: 1) Accumulation of unresolved decisions, unimplemented actions, and compromises made by leaders [33–35]. 2) Outdated organisational structures, policies and processes that no longer serve the organisation's objectives [23, 30]. 3) The gap between the organisation's strategic plans and actual capacity to implement them [25, 26]; and 4) Interest paid in the form of reduced efficiency, innovation, and adaptability [32, 39]. Synthesizing these recurring concepts, this study defines organisational debt as:

*" Organizational debt is the accumulation of outdated structures, policies, and processes that are no longer advantageous for the organization. Consequently, obstructs the advancement of the organization and its capacity to adjust to evolving conditions, ultimately hindering its potential for excellence."*

This concept encompasses the fundamental aspects of OD that have arisen from the literature. Firstly, it appears in outdated organizational elements that do not correspond with strategic goals. Furthermore, it hinders the advancement of the organization and its ability to adjust to new circumstances. This succinct yet all-encompassing description establishes a conceptual basis for discussing the origins, effects, and administration of OD.

## 4.3 Causes and consequences of organizational debt

The MLR identified multiple primary factors contributing to the accrual of OD. These causes indicate inadequacies in leadership, culture, teamwork, and change management. Table 2 summarizes the key causes to and consequences of OD:

**Table 1. OD definitions.**

| Proposed Definition | Year | Source |
|---|---|---|
| "OD is all the people/culture compromises made to 'just get it done' in the early stages of a start-up" | 2015 | [35] |
| "Organizational debt is any structure or policy that no longer serves an organization" | 2020 | [30] |
| "Organizational debt is the accumulation of changes and decisions leaders should have made but did not" | 2016 | [33] |
| "The interest companies pay when their structure & policies stay fixed and/or accumulate as the world changes" | 2016 | [32] |
| "Management Debt is incurred when you make an expedient, short-term management decision with an expensive, long-term consequence" | 2022 | [24] |
| "Organizations may intentionally or unintentionally incur organizational debt through management actions, governance process changes, internal process changes, or large-scale organizational changes when short-term advantages are sought at the expense of 'doing things right'" | 2017 | [36] |
| "Organizational debt is the baggage that prevents people from delivering astonishing results" | 2015 | [28] |
| "Organizational debt—our organizations are also a good excuse to avoid changes, as we often look for someone who is going to help us, but we do not really want to give him or her the power to implement the changes" | 2019 | [29] |
| "Organizational debt is sibling of technical debt, for example a toxic culture, struggling leader etc." | 2020 | [27] |
| "Organizational debt: things that should've been done to ensure health & efficiency, but weren't" | 2021 | [41] |
| "Organizational debt, an analogy! During the execution of organizational changes (transformations, reorganizations, changes in ways of working etc.) shortcuts are taken that lead to frustration, more time and money etc. It's the same thing as technical debt" | 2021 | [23] |
| "Organizational Debt is the interest companies pay when their structure and policies 1) stay fixed and/or 2) accumulate as the world changes" | 2016 | [39] |
| "Organizational debt is a holistic concept, and it is more than technical debt and also different from bureaucracy. Organizational debt is a networked concept that fosters the blame-free identification of cross-functional and cross-department weak points" | 2023 | [43] |

**Table 2. OD causes and consequences.**

| Cause of OD | Consequences of OD |
|---|---|
| Poorly managed organizational change | Reduced efficiency, productivity, innovation and morale |
| Lack of collaboration culture | Lack of agility to adapt to market changes |
| Siloed change efforts | Reduced competitiveness and stagnating growth |
| Avoiding confrontation of issues | Entrenched bureaucracy and resistance to change |

*Poorly Managed Organizational Change*: Undertaking significant modifications without sufficient preparation, instruction, and involvement of relevant parties frequently results in the failure to attain the intended results. Insufficient communication of the rationale and benefits of introducing new tools or processes can lead to confusion among staff. Inadequate training on new systems leaves employees struggling to adapt, reducing productivity and morale. Failure to involve key stakeholders early in the change initiative overlooks valuable insights and acceptance. Poor change management spreads dissatisfaction and resistance, distracting focus from regular operations. Ultimately, the change initiative falls short of goals and disrupts workflows. This poorly managed change incurs OD through reduced efficiency, disengaged staff who resent imposed changes, and lack of flexibility to keep pace with business needs [31, 37].

*Lack of Collaboration Culture*: Excluding staff from decision-making and change initiatives overlooks valuable perspectives and insights from those closest to the work. A lack of openness to bottom-up input and ideas hinders innovation. Silos between teams or departments prevent

knowledge sharing which can highlight process inefficiencies and redundancies. Leaders who resist input from below risk disenfranchising employees, reducing engagement and motivation. Poor collaboration limits the organization's ability to leverage its collective intelligence fully. Instead of synergistic workflow, fragmented efforts reduce process efficiency, agility, and continuous improvement [36].

*Siloed Change Efforts*: When different departments or teams undertake changes in isolation, this leads to duplicated efforts, incompatible systems, and wasted resources. A lack of communication and coordination between disparate change initiatives produces fragmented outcomes. Divergent tools, policies, and processes increase complexity across the organization. This results in decreased productivity, since efforts to resolve misalignments and conflicts between departments are diverted from productive endeavors. Additionally, redundant endeavors signify neglected prospects to maximize the utilization of communal resources and expertise. Without an integrated approach to change management, inferior outcomes are inevitable [31, 36].

*Avoiding Discontent*: A reluctance to confront subpar performance and opposition to essential modifications contribute to the continuation of inefficiency. This acceptance of mediocrity becomes the norm when supervisors are unable to approach troublesome employees for fear of retaliation. When high achievers passively labor with underachievers, they are burdened. By evading necessary yet disruptive adjustments for fear of inciting complaints, one can perpetuate antiquated methods and misalignments. Although the immediate benefit may be to preserve peace, the long-term repercussions of continued mediocrity and stagnation are considerably more severe. An organization incurs debt as challenging matters continue to accrue unsolved [33, 34]. These root causes interact to produce a range of detrimental consequences for software organizations:

*Reduced efficiency*, productivity, innovation and morale: diminished efficacy, output, ingenuity, and team spirit [28, 32]. Displeased employees perceive their contributions as underappreciated and their time squandered on repetitive tasks or navigating complex administrative procedures.

*Lack of agility* and inability to adapt to market changes: insufficient flexibility and incapacity to adjust to shifts in the market [25, 26]. The organization's outdated legacy systems and bloated processes hinder its ability to stay up with more agile competitors, making it slow and inflexible.

*Reduced competitiveness and stagnant growth* [25, 29]. The presence of TD and inefficient procedures necessitates a significant allocation of resources, which in turn hampers the ability to invest substantially in innovation.

The presence of a deeply rooted administrative system and opposition to altering it [25, 34]. The accumulation of organizational complexity persists as people adhere to familiar heritage systems. Efforts to initiate change struggle as employees have a sense of powerlessness.

Therefore, OD incurs major "interest" costs from reduced productivity, innovation lag, technology debt, opportunity costs from forfeited growth, and an increasingly dysfunctional workplace. Software leaders must recognize this self-reinforcing downward spiral and intervene promptly to restore organizational fitness and strategic alignment.

## 4.4 Identification and mitigation of organisational debt

Identifying and mitigating organisational debt requires a systematic approach given its multifaceted nature. Organisational debt symptoms can be spotted through regular performance monitoring, employee surveys, and audits of processes [38, 41]. Comprehensive evaluation of various organisational components is vital for deeper insights [40].

Quantitative performance metrics offer warning signs such as prolonged declines in productivity, increasing software defects, lags in new feature release, product quality issues, and rising customer complaints. Comparing metrics over time and against competitors highlights underperformance. Periodic audits help assess process efficiency, redundancy, and alignment with objectives. Surveys and interviews to gather employee perspectives on pain points complement the top-down analysis. The utilization of both quantitative and qualitative data allows for the cross-validation of findings regarding the state of organizational components.

Signs of insufficient regulations, protocols, and organizational norms encompass role ambiguity, decision-making barriers, unbalanced incentives, restricted openness, and isolated information. Process debt can be identified through the confluence of static production metrics and employee discontentment stemming from an overabundance of bureaucratic procedures. The TD resulting from the deterioration of product quality, as evidenced by developers' grievances regarding impracticable deadlines, underscores the need for a reassessment of administrative objectives. Through this methodical evaluation of symptoms, problem areas that necessitate improvement are identified.

By placing value delivery above hierarchical structures and control processes, software teams may concentrate on rapidly satisfying stakeholder requirements with shippable increments. The adoption of agile concepts, including the establishment of autonomous, cross-functional, and small teams, serves to mitigate the occurrence of isolated tasks and empowers personnel to devise strategies and execute them autonomously. Minimally feasible procedures provide an adequate degree of structure to facilitate outcomes, while avoiding the development of too intricate bureaucratic systems. Complete visibility of the status of work is achieved through the utilization of information radiators, boards, and charts, hence augmenting transparency. This promotes confidence, shared accountability, and reciprocal liability for results. Efficient procedures that prioritize initial advantages foster innovation, adaptability, and the capacity to maintain competitiveness in the face of adversity [2, 3].

Stagnation can be reduced by fostering role creation and using flexible work practices, which enable people to customize their responsibilities in accordance with their skill sets. Micromanagement has a detrimental effect on employee motivation since it communicates a deficiency in confidence and recognition of their abilities, which in turn stifles the innovative thinking required to resolve complex problems. Providing teams with the autonomy to devise their own strategies for attaining mutually agreed-upon objectives enhances their degree of engagement. When individuals avoid from erecting hard demarcations between positions, they are better able to readily adapt to changing circumstances and respond with adaptability to emerging issues. It is possible to cultivate a culture of innovation inside an organization that fosters psychological safety by encouraging individuals to take risks and gaining important insights from failures. It is possible to encourage the innate motivation of individuals and foster flexibility by offering deliberate autonomy to individuals rather than enforcing strict conformance to customary processes [2].

Continuous feedback from frontline personnel provides timely insights into inconsistencies that may occur between stated protocols and practical scenarios. These inconsistencies may appear in a variety of situations. Traditional surveys that are anonymous are able to obtain honest responses without the risk of negative consequences. Town hall meetings create an atmosphere that encourages the free and open airing of feelings of dissatisfaction and the sharing of potential solutions. Those in leadership positions who aggressively seek criticism demonstrate a willingness to receive input and a dedication to working together to find solutions to problems. This encourages symbiotic relationships with both employees and management, as opposed to transactional interactions that are more likely to be characterized by mistrust. Opportunities to modify systems in a more effective manner that promotes human flourishing

can be discovered through a critical evaluation of disparities between objectives and results [2].

By involving employees in decentralized and participatory decision-making, policies and processes may be flexibly adapted to evolving realities. Absent extensive stakeholder input, systems intended for abstraction frequently deviate from requirements. Nobody comprehends uncertainties and sources of friction more thoroughly than personnel who are immersed on a daily basis. The insights they possess are often overlooked by questionnaires due to their personal experiences. Architectures based on collaborative choice and co-creation foster psychological investment and alignment. Individuals performing the tasks are in the best position to engage in open problem-solving as obstacles arise via cycles of experimentation, feedback, and learning [2, 3].

Monitoring metrics on performance, behavior, and innovation provides a holistic dashboard. Productivity goals have potential to dehumanize work into joyless drudgery if not complemented by gauges of employee experience, health, and ability to creatively contribute. Well-rounded assessments consider both business and human needs. For instance, software teams require sufficient time for refactoring TD to maintain velocity long-term. Quality cannot be sacrificed for feature output volume. Regular pulse checks on morale, trust, purpose, and work-life balance offer leading indicators before burnout [11].

Implementing programs that allow frontline employees to anonymously identify mismatched policies helps prevent the divergence of systems from actual demands. Providing incentives for constructive critiques encourages the identification of defective processes at an early stage, thus preventing the accumulation of debt. Encouraging disagreement exhibits a sense of psychological security to question the existing state of affairs. Leaders must acknowledge that policies are not permanent and need to be regularly reassessed as circumstances change. The key differentiating factor between learning organizations and stagnant bureaucracies is the ongoing evaluation of whether current structures and norms effectively meet the evolving needs. This process of continuous reflection is emphasized in learning organizations [2, 56].

In summary, continuously tuning processes and structures requires decentralizing power, trusting employees, and weighing quantitative metrics against qualitative insights. An agile mindset focused on early delivery of value, adaptation in response to learning, and leveraging collective intelligence helps curb debt. Leaders play a key role in diagnosing and addressing sources of debt through organizational redesign. However, they often lack deep visibility into the causes of inertia, misalignment, and inefficiency on the frontlines. Engaging staff in open discussions to uncover pain points is invaluable, combined with a culture of psychological safety where people feel secure speaking up. Dismantling OD requires placing human needs on par with business needs.

### 4.5 Future research directions

While this literature review has consolidated understanding of OD, several fruitful avenues exist for further investigation based on current knowledge gaps:

- Develop metrics to quantify OD, enabling rigorous tracking and benchmarking. Combine productivity data with indicators of culture, innovation, and TD [52]. Establish validated scales to measure dimensions like employee engagement, psychological safety, organizational agility, and leadership effectiveness [57–59]. Statistical modeling can relate these metrics to OD.

- Conduct empirical studies on the impact of OD on workforce motivation, attrition, fatigue, and burnout [60–62]. Use questionnaires and ethnographic methods to gather insights.

Relate debt to tangible individual performance metrics like productivity, absenteeism, and error rates.

- Investigate through case studies the relationship between OD and customer satisfaction [63, 64], especially in software-intensive service organizations. Survey data can correlate debt to metrics like call resolution times, complaint rates, churn, and net promoter scores.

- Examine through controlled experiments the role of OD in software project success/failure [65–67]. Vary team structures and processes to reveal optimal configurations. Productivity, quality, cost, and schedule metrics assess performance.

- Estimate the economic costs of OD through case studies and cost-modeling across software companies [68–70]. Assess opportunity costs from delayed innovations. Relate to total cost of ownership models.

- Explore whether TD quantification techniques [71–73] can be extended to provide estimates of OD. Compare their accuracy and utility.

- Design field studies of interventions such as restructured teams, revised workflows, and new planning processes to validate OD mitigation techniques [74–76]. Measure before and after effects.

Further research to address these gaps will provide more rigorous, empirically-grounded insights to guide debt management in practice. It represents an emerging interdisciplinary arena spanning management science, organizational behavior, anthropology, and software engineering [77]. Collaboration between academics and industry practitioners is needed to develop context-specific strategies rooted in both theory and pragmatism [78]. There are rich possibilities for cross-pollination between disciplines to uncover novel solutions [79]. With organizational agility and adaptability growing ever more crucial in turbulent conditions [80], understanding how to minimize friction and debt represents the key to sustaining innovation and competitiveness [56].

## 5 Validity threats and methodological limitations

This section examines the constraints of the study design and any factors that could undermine the accuracy and reliability of its results. This MLR follows the methodological framework proposed by Garousi and Mantyla [21]. It conducts a thorough assessment of the external, internal, and construct validity, while also recognizing the inherent limits of the research process.

*External Validity Threat*: One of the significant risks is associated with the lack of the peer-review process in the practitioner literature. Grey literature offers practical information that is acquired from an industrial setting. Still, such absence of a broad empirical basis warns of a potential threat of built-in bias. The contextual details of the described experiences are not clear. As such, the findings cannot be generalized to software enterprises at large. This study strengthening the results with more analysis from studies published in peer-reviewed journals to support the conclusions. In addition, the limitation to English primary sources raises the concern about the degree of accessibility within other areas of language. Despite this drawback, it is reasonable to state that MLR covers a large body of information, thus facilitating the representation of the views of both TD and NTD experts in a fully adequate manner.

*Internal validity Threat* pertains to the evaluation of potential biases that may influence the outcomes of the MLR analysis. The primary objective is to identify and mitigate these biases. Employing a pre-established search engine, specific search phrases, and well-defined

inclusion/exclusion criteria as components of a systematic source selection process is a dependable strategy to ensure consistent results. However, following careful analysis, one can detect inherent dangers linked to limitations on search terms, reliance on a solitary search engine, and potential biases in the execution of inclusion/exclusion criteria. To tackle these problems, a structured search was carried out using precise keywords in a methodical fashion, aiming to reduce the chance of overlooking pertinent studies. The research team, comprising two seasoned software engineering researchers, and a third researcher to be an auditor bolsters internal validity by using their extensive expertise and educational qualifications, thereby mitigating potential risks to some extent.

*Construct Validity Threat*: One of the major challenges to construct validity is the absence of empirical evidence in primary research. The dependence on gray literature, which mostly consists of subjective opinions and practical knowledge from professionals, poses a possible risk. Although this type of information provides a significant insight into real-world industrial situations, the lack of specific contextual elements raises an epistemological question about the clarity of the knowledge source [21]. However, it is important to acknowledge that this constraint, arising from the characteristics of grey literature, does not automatically render the findings erroneous. A proposed solution is envisioning a situation where the identical professionals, who have expressed their viewpoints in unofficial publications, do structured interviews or surveys on the identical topic. Adopting such a strategy would probably produce reliable outcomes, however with an increased level of methodological precision.

The use of blog posts and online articles inherits further validity threats. The expertise and qualifications of authors are difficult to ascertain. Sample size was small, constrained to only English search results. Relevant publications in other languages were excluded. The anecdotal nature of this literature reduces reliability compared to large-sample studies. However, the findings still offer useful preliminary insights on an emerging topic with minimal academic attention currently. Despite limitations, this exploratory MLR serves the intended purpose of mapping key concepts, issues, and questions on organisational debt to guide future research.

To summarize, the MLR effectively deals with validity threats and limitations, providing valuable insights into the complex field of software engineering challenges. However, it is crucial to continuously consider the methodology and prioritize transparency in order to improve research efforts in this area.

## 6 Discussion

This study identifies OD as a complex problem, closely connected to factors such as corporate culture, leadership dynamics, collaborative frameworks, process design complexities, decision-making paradigms, and organizational structures. This study synthesized practitioner and academic perspectives to clarify the complex phenomenon of organizational debt (OD) within the software engineering context. By integrating insights from grey literature and peer-reviewed sources, our analysis revealed both convergent and divergent viewpoints, enriching our understanding of OD dynamics.

### 6.1 Convergence of perspectives

Both practitioner and academic sources acknowledged the adverse consequences of OD, including reduced productivity, innovation, and competitiveness [28, 32, 45, 48, 49]. The accumulation of outdated structures, policies, and processes was consistently identified as a root cause, reflecting the organization's inability to adapt to changing environments [23, 30, 33, 45, 48, 49]. Furthermore, there was a consensus on the detrimental impact of siloed operations,

lack of cross-functional collaboration, and resistance to change, which fuel the growth of OD [30, 35, 51, 79].

## 6.2 Divergence of perspectives

While practitioner sources emphasized the role of leadership decisions, compromises, and avoidance of disruptive changes in contributing to OD [33–35], academic literature investigated the organizational and human factors. Aspects such as rigid hierarchies, ineffective coordination mechanisms, technostress, and employee disengagement were highlighted as significant OD drivers [46, 50, 52, 53]. This divergence underscores the multifaceted nature of OD, which extends beyond individual decision-making to encompass systemic organizational and technological complexities.

## 6.3 Implications for software engineering research

The integration of practitioner and academic viewpoints opens up avenues for future research in software engineering. Empirical investigations into the interplay between OD, software project outcomes, and organizational performance are warranted. Quantitative studies could explore the economic impact of OD, complementing existing technical debt quantification techniques [71–73]. Qualitative inquiries could unravel the socio-technical dynamics underpinning OD accumulation and its relationship with organizational culture, leadership, and team dynamics.

Furthermore, interdisciplinary collaborations with management science, organizational behavior, and information systems researchers hold promise for cross-pollinating insights and developing holistic OD management frameworks tailored to software-intensive organizations. Longitudinal studies tracking the effectiveness of interventions, such as restructuring, process redesign, and cultural transformation, could validate proposed mitigation strategies empirically.

## 6.4 Implications for software engineering practice

This study's findings underscore the importance of proactive OD management for software organizations. Practitioners should prioritize continuous assessment and adaptation, fostering organizational agility and flexibility [48, 49, 51]. Regular audits, employee feedback, and performance monitoring can aid in identifying OD hotspots and informing timely interventions.

Aligning organizational structures, processes, and strategic objectives is crucial, while considering the long-term implications of decisions [47, 48]. Effective technology governance, digital upskilling initiatives, and addressing technostress can mitigate technology-related OD drivers [50, 53].

Moreover, empowering employees, promoting a learning culture, and encouraging cross-functional collaboration can strengthen an organization's collective intelligence and capacity for innovation [49, 50]. Leaders should embrace decentralized decision-making, bottom-up ideation, and employee autonomy to foster an engaged and adaptable workforce [2, 3, 36, 46].

## 6.5 Discussion takeaway

This study, conceived as the first step of a community-led effort, is in line with the agile philosophy, promoting cooperation, adaptation, and ongoing improvement to collaboratively tackle the difficulties presented by OD.

In summary, OD can be defined as the comprehensive and nuanced characterization resulting from the synthesis of 13 diverse definitions of OD collected from the literature;

"*Organizational debt is the accumulation of outdated structures, policies, and processes that are no longer advantageous for the organization. Consequently, obstructs the advancement of the organization and its capacity to adjust to evolving conditions, ultimately hindering its potential for excellence.*"

*Manifestations and Impact of OD*: Arising from the dual forces of obsolescence and accumulation, OD shares discernible parallels with process debt [19, 81]. The deleterious impact of OD resonates within the workforce, manifesting as a detriment to employee morale and performance. Developers, burdened by legacy issues stemming from OD, grapple with disengagement, hindering their ability to focus on innovation and creative problem-solving. Embracing the core tenets of the Agile Manifesto becomes imperative to infuse flexibility and adaptability, averting the pitfalls of superficial or 'fake' agile implementations that may lead to escalated costs, delays, and pervasive frustration.

*Mitigation Strategies for OD*: Effectively navigating the labyrinth of OD mitigation mandates a holistic approach. Proactive identification and mitigation of OD sources emerge as linchpins for bolstering adaptability, efficiency, and sustained success. Acknowledging the inevitability of debt accumulation during organizational growth, leaders assume the role of diagnosticians, continuously evaluating areas where obsolescence or accumulation begets challenges [24, 33, 34]. The evaluative lens extends to the organizational structure, demanding continuous refinement to ensure alignment with strategic objectives and responsiveness to change. Adopting established TD quantification techniques represents a starting point for addressing OD. However, the social and organizational complexities of OD may require customized mitigation practices based on contextual needs.

*Agile Mindset and Technological Adaptation*: Championing an agile mindset serves as the catalyst for fostering resilience and innovation, propelling organizations to confront the ever-evolving business landscape. A flexible work environment, underpinned by a culture valuing innovation, experimentation and change, becomes pivotal. The cultivation of a continuous improvement ethos and the integration of feedback mechanisms empower employees, fortifying organizational agility. While performance, behavioral, and innovation metrics provide valuable insights into OD [42], their efficacy hinges upon contextual considerations within the agile paradigm.

*Embracing Technological Advancements*: Within the agile setting, embracing technological advancements emerges as a fundamental strategy for OD mitigation. Companies that stay updated on the latest technology and strategically incorporate them into their operations have simpler procedures and increased efficiency. This progressive strategy is in perfect harmony with agile principles, prioritizing quick adjustment to new tools and processes for continuous enhancement.

*Optimizing Organizational Structures*: It becomes crucial to prevent the accumulation of unneeded complications, which requires the optimization of organizational structures. An environment that fosters ongoing enhancement, along with a proactive attitude towards embracing change as a means of progress, establishes the foundation for a software company that is more adaptable and robust.

*Future Directions and Community-Driven Initiatives*: This innovative study, utilizing MLR approaches, examines the causes and effects of organizational dysfunction, represented as OD in software products and professional services. To tackle these difficulties, it is essential for academia and software firms to work together, following the agile philosophy of cross-functional teamwork. Adopting known TD methods [8, 9] serves as a fundamental approach to OD. Nevertheless, the distinct difficulties presented by OD necessitate the development of innovative approaches for its resolution. This study, conceived as the first step of a community-led effort, is in line with the agile philosophy, promoting cooperation, adaptation, and ongoing improvement to collaboratively tackle the difficulties presented by OD.

## 7 Conclusion

This MLR offers a conceptual foundation for the nascent topic of OD in software engineering, synthesizing diverse perspectives from industry experts. The analysis reveals OD arises from the accumulation of outdated structures, policies, and processes that are misaligned with an organization's strategic goals. Poor change management, siloed efforts, avoiding disruption, and resistance to input are identified as key debt drivers. Consequences encompass reduced innovation, agility, competitiveness, morale, and employee retention.

Mitigating OD necessitates embracing agile principles such as cross-functional teams, minimum viable processes, transparency, decentralization, and delivering value rapidly. Ongoing feedback mechanisms, participatory decision-making, and continuous alignment of systems to evolving needs are vital. However, the reliance on non-peer-reviewed practitioner sources raises validity concerns regarding the lack of rigorous empirical grounding. Further research across diverse organizational and team settings is critically needed to validate the suggested OD causes, impacts, metrics, and management techniques.

Developing practical solutions requires cooperative efforts between academic institutions and software companies. OD management can build on existing best practices for quantifying, prioritizing and repaying TD, while also accounting for the distinct socio-technical complexities of OD. Further studies should investigate measurable metrics, workforce effects like burnout and turnover, customer satisfaction impacts, project success/failure outcomes, economic costs, and tailored organizational interventions.

Addressing this emerging interdisciplinary research domain necessitates drawing from diverse fields including management science, organizational behavior, anthropology, network analysis, and software engineering. Unraveling the complexities of OD is the key to maintaining innovation in the face of complexity and uncertainty. This is because organizational agility and flexibility are becoming increasingly important for competitiveness. To effectively address this challenge, a synergistic partnership between the business world and the academic world is required.

By synthesizing practitioner and academic insights, this study provides a comprehensive understanding of OD, emphasizing its multidimensional nature and the need for holistic management approaches. Software organizations can leverage these findings to diagnose OD sources, mitigate adverse consequences, and cultivate an agile and innovative organizational culture, ultimately enhancing their competitiveness and long-term success.

The conclusions from this exploratory MLR establish a conceptual foundation and future research agenda for the novel phenomenon of OD. Rigorously validating the identified concepts in empirical studies across diverse settings remains an open avenue. Developing contextualized management strategies demands a human-centric approach considering both business performance and employee experience metrics. As software pervades all facets of life, understanding the obstacles posed by dysfunctional structures and policies is crucial for organizational excellence.

## Supporting information

**S1 Checklist. PRISMA checklist.**
(DOCX)

## Author Contributions

**Conceptualization:** Osama Al-Baik, Mwaffaq Abu Alhija, Muhammad Ovais Ahmad.

**Data curation:** Osama Al-Baik, Mwaffaq Abu Alhija, Muhammad Ovais Ahmad.

**Formal analysis:** Osama Al-Baik, Mwaffaq Abu Alhija, Muhammad Ovais Ahmad.

**Investigation:** Osama Al-Baik, Mwaffaq Abu Alhija, Muhammad Ovais Ahmad.

**Methodology:** Osama Al-Baik, Mwaffaq Abu Alhija, Hikmat Abdeljaber, Muhammad Ovais Ahmad.

**Resources:** Hikmat Abdeljaber, Muhammad Ovais Ahmad.

**Validation:** Mwaffaq Abu Alhija, Hikmat Abdeljaber, Muhammad Ovais Ahmad.

**Writing – original draft:** Osama Al-Baik, Mwaffaq Abu Alhija, Muhammad Ovais Ahmad.

**Writing – review & editing:** Osama Al-Baik, Mwaffaq Abu Alhija, Hikmat Abdeljaber, Muhammad Ovais Ahmad.

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
