## [Decision Letter · Decision Letter 0]

22 May 2024

PONE-D-24-05123Organizational Debt – Roadblock to Agility in Software Engineering: Exploring an Emerging Concept and Future Research for Software ExcellencePLOS ONE

Dear Dr. Al-Baik,

Thank you for submitting your manuscript to PLOS ONE. After careful consideration, we feel that it has merit but does not fully meet PLOS ONE’s publication criteria as it currently stands. Therefore, we invite you to submit a revised version of the manuscript that addresses the points raised during the review process.

**Please note that we have only been able to secure a single reviewer to assess your manuscript. We are issuing a decision on your manuscript at this point to prevent further delays in the evaluation of your manuscript. Please be aware that the editor who handles your revised manuscript might find it necessary to invite additional reviewers to assess this work once the revised manuscript is submitted. However, we will aim to proceed on the basis of this single review if possible. **

We look forward to receiving your revised manuscript.

Kind regards,

Vanessa Carels

Staff Editor

PLOS ONE

Journal Requirements:

Reviewers' comments:

Reviewer's Responses to Questions

**Comments to the Author**

1. Is the manuscript technically sound, and do the data support the conclusions?

Reviewer #1: Partly

2. Has the statistical analysis been performed appropriately and rigorously? 

Reviewer #1: I Don't Know

3. Have the authors made all data underlying the findings in their manuscript fully available?

Reviewer #1: No

4. Is the manuscript presented in an intelligible fashion and written in standard English?

Reviewer #1: Yes

5. Review Comments to the Author

**Reviewer #1: **Overall, the study addresses an interesting topic that worth investigating. It well explained the motivation and purpose for the study, and use the right methodology to approach the study.

However, the main limitation of the study is the lack of data to sufficiently ground the findings. The data is only contain a few amount of blogs (22 blogs) with no idea how intense each blog is. This limit amount of data might severely undermine the quality of the results. I would suggest the authors to augment the data with some related research papers about OD. You might need to go beyond software engineering research avenue (checking research avenue in the management and organization area) , examples are:

- William F. Jarvis. "Organizational debt levels: harbinger of change? Among healthcare organizations, debt has increased steadily in recent years. Now, most are deleveraging--perhaps in response to less favorable operating trends"

- Liu, Zixuan, Viktoria Stray, and Tor Sporsem. "Organizational Debt in Large-Scale Hybrid Agile Software Development: A Case Study on Coordination Mechanisms."

6. PLOS authors have the option to publish the peer review history of their article (what does this mean?). If published, this will include your full peer review and any attached files.

Reviewer #1: No

---

## [Author Response · Author response to Decision Letter 0]

26 Jun 2024

Dear Editors and Reviewers,

Thank you for your valuable feedback and suggestions on our manuscript titled "Organizational Debt in Software Engineering: A Multivocal Literature Review." We appreciate the time and effort you have invested in reviewing our work and providing constructive comments to improve its quality.

We have carefully considered the points raised by the reviewer and would like to address them in this response letter.

Regarding the concern about the lack of data to sufficiently ground the findings, we acknowledge the limitation of relying solely on blog posts and practitioner literature. As mentioned in the paper, the topic of organizational debt (OD) is an emerging concept in software engineering, and there is a scarcity of academic literature on this subject. Our motivation for conducting a multivocal literature review (MLR) was to synthesize the perspectives and experiences of industry professionals, who are at the forefront of grappling with OD challenges.

However, we recognize the importance of complementing these practitioner insights with academic research to strengthen the validity and rigor of our findings. We thank the reviewer for suggesting the paper by William F. Jarvis and the case study by Zixuan Liu et al. These sources provide valuable insights from the management and organization domains, which can enrich our understanding of OD in the software engineering context.

To address this concern, we propose the following revisions to our manuscript:

1. We conducted a thorough search for relevant academic literature on OD from related disciplines, such as management, organizational behavior, and information systems. This has broaden our data corpus and has provided a more comprehensive view of the phenomenon. Please, see pages 4-6

2. We incorporated the suggested papers by William F. Jarvis and Zixuan Liu et al., as well as eight (8) other relevant academic sources, into our analysis. This allows us to triangulate the practitioner perspectives with theoretical and empirical findings from academic research. Please, see results section on pages 5-6.

3. In the revised manuscript, we clearly distinguish between insights derived from practitioner literature and those from academic sources. This will hopefuly enhance transparency and will allow the reader to assess the credibility and generalizability of our findings. Please, see results section on pages 5-6.

4. We adjusted the methodology section to reflect the inclusion of academic literature and provide a detailed description of the search strategy, selection criteria, and analysis process for these additional sources. Please, see research method section on pages 4-5.

5. The discussion section has been expanded to critically evaluate the convergence or divergence of practitioner and academic perspectives on OD, as well as the implications for software engineering research and practice. Please, see discussion section on pages 13-14.

We believe that addressing the reviewer's concern by incorporating relevant academic literature has strengthen our study's validity and contribute to a more comprehensive understanding of the organizational debt phenomenon in software engineering.

Thank you again for your valuable feedback and the opportunity to improve our work. We look forward to addressing these concerns and submitting a revised manuscript for your consideration.

Sincerely,

Osama Al-Baik

---

## [Decision Letter · Decision Letter 1]

18 Jul 2024

Organizational Debt – Roadblock to Agility in Software Engineering: Exploring an Emerging Concept and Future Research for Software Excellence

PONE-D-24-05123R1

Dear Dr. Osama Al-Baik,

We’re pleased to inform you that your manuscript has been judged scientifically suitable for publication and will be formally accepted for publication once it meets all outstanding technical requirements.

Kind regards,

Leander Luiz Klein, Ph.D.

Academic Editor

PLOS ONE

Additional Editor Comments (optional):

Dear authors!

Thank you for making the revision of the article following the reviewer' suggestion. I send the article to the editor in chief with my final decision.

Best regards.

Reviewers' comments:

Reviewer's Responses to Questions

**Comments to the Author**

1. If the authors have adequately addressed your comments raised in a previous round of review and you feel that this manuscript is now acceptable for publication, you may indicate that here to bypass the “Comments to the Author” section, enter your conflict of interest statement in the “Confidential to Editor” section, and submit your "Accept" recommendation.

Reviewer #1: All comments have been addressed

2. Is the manuscript technically sound, and do the data support the conclusions?

Reviewer #1: Yes

3. Has the statistical analysis been performed appropriately and rigorously? 

Reviewer #1: Yes

4. Have the authors made all data underlying the findings in their manuscript fully available?

Reviewer #1: Yes

5. Is the manuscript presented in an intelligible fashion and written in standard English?

Reviewer #1: Yes

6. Review Comments to the Author

Reviewer #1: (No Response)

7. PLOS authors have the option to publish the peer review history of their article (what does this mean?). If published, this will include your full peer review and any attached files.

Reviewer #1: **Yes: **Abdullah Aldaeej

---

## [Editor Report · Acceptance letter]

28 Oct 2024

PONE-D-24-05123R1 

PLOS ONE

Dear Dr. Al-Baik, 

I'm pleased to inform you that your manuscript has been deemed suitable for publication in PLOS ONE. Congratulations! Your manuscript is now being handed over to our production team.

Kind regards, 

on behalf of

Professor Leander Luiz Klein 

Academic Editor

PLOS ONE